# Soft Tissue Mobilization and Stretching for Shoulder in CrossFitters: A Randomized Pilot Study

**DOI:** 10.3390/ijerph18020575

**Published:** 2021-01-12

**Authors:** Marcos Jusdado-García, Rubén Cuesta-Barriuso

**Affiliations:** Department of Physiotherapy, Faculty of Sport Sciences, European University of Madrid, 28670 Madrid, Spain; marcosgarjus@gmail.com

**Keywords:** instrument-assisted soft tissue mobilization, muscle stretching exercises, range of motion, manual therapy

## Abstract

Background. The shoulder in CrossFit should have a balance between mobility and stability. Glenohumeral internal rotation deficit and posterior shoulder stiffness are risk factors for overhead shoulder injury. Objective. To determine the effectiveness of instrument-assisted soft tissue mobilization and horizontal adduction stretch in CrossFit practitioners’ shoulders. Methods: Twenty-one regular CrossFitters were allocated to experimental (stretching with isometric contraction and instrument-assisted soft tissue mobilization) or control groups (instrument-assisted soft tissue mobilization). Each session lasted 5 min, 2 days a week, over a period of 4 weeks. Shoulder internal rotation and horizontal adduction (digital inclinometer), as well as posterior shoulder stretch perception (Park scale), were evaluated. Shapiro–Wilk test was used to analyze the distribution of the sample. Parametric Student’s *t*-test was used to obtain the intragroup differences. The inter- and intra-rater differences were calculated using a repeated measures analysis of variance (ANOVA). Results. Average age was 30.81 years (SD: 5.35), with an average height of 178 (SD: 7.93) cm and average weight of 82.69 (SD: 10.82) kg. Changes were found in the experimental group following intervention (*p* < 0.05), and when comparing baseline and follow-up assessments (*p* < 0.05) in all variables. Significant differences were found in the control group following intervention (*p* < 0.05), in right horizontal adduction and left internal rotation. When comparing the perception of internal rotation and horizontal adduction in both groups, significant differences were found. Conclusions. Instrument-assisted soft tissue mobilization can improve shoulder horizontal adduction and internal rotation. An instrument-assisted soft tissue mobilization technique yields the same results alone as those achieved in combination with post-isometric stretch with shoulder adduction.

## 1. Introduction

CrossFit is a physical fitness system featuring the performance of a wide variety of exercises covering sports disciplines (weightlifting, powerlifting, and gymnastics) in addition to activities such as running, rowing, or cycling. Workouts are combined with little or no rest, involving high-intensity training [1].

With regard to the incidence of injuries in the practice of CrossFit, there is a scarce amount of data published in the literature [2], with an estimated rate of 3.1 injuries per 1000 h of training. This prevalence is similar to that found in sports such as weightlifting, gymnastics, and rugby (3–3.3/1000 h). The prevalence of musculoskeletal injury was 24.0%, and the most affected regions of the body were the lumbar spine, shoulders, and knees [3].

CrossFit is an overhead sport, in which many of the movements are performed above the head, as in other sports such as baseball, volleyball, or tennis. However, unlike these, the burden does not rest exclusively on the dominant upper limb, but is shared between the two extremities [3]. This sport requires a sufficiently lax shoulder to be able to reach extreme positions of movement above the head, but with enough stability to prevent luxation. CrossFit performance is thus associated with different power-, strength-, and aerobic-related markers [4]. In most CrossFit exercises, athletes not only have to lift or throw an external load, but also their own body mass. For this reason—as in other sports—trying to reach a balance between maximum strength and body mass will be of paramount importance [5].

Training more than four days a week and not receiving regular physiotherapeutic care were associated with CrossFit-related musculoskeletal injuries [3]. The high incidence of injury in the shoulder joint is due to various etiological factors. In Olympic lifting (typically in weightlifting) and gymnastics movements, the shoulders need to reach extreme positions of flexion, adduction, external rotation, and internal shoulder and elbow extension is required. These movements occur when the head is placed under the bar in Olympic lifts and kipping pull-ups in gymnastics, in this case, using the moment of inertia below the bar in the performance of chin-ups or similar exercises [1]. These movements are performed through a series with long reps and using large weights at high intensity, which can lead to muscle fatigue, poor technique, and an alteration in shoulder joint alignment [6].

The stability of the glenohumeral joint depends, to a great extent, on its active stability. Muscle fatigue, caused by repetitive high-intensity exercises in CrossFit, can have a detrimental effect on the activity and muscle response in these athletes. This muscle alteration caused by fatigue produces a decrease of the dynamic joint stability; a poorer technique; and, as a result, a greater likelihood of injury [1].

Most injuries in CrossFit occur as a result of repetitive strain, implying an extended process in time, which can lead to a higher prevalence of injury. Athletes acquire adaptations from the sport itself, including alterations in strength, flexibility, and posture, which induce changes in the biomechanics and movement patterns [7]. Therefore, overhead athletes are participant to the risk of injury in the shoulder joint due to overuse, such as deficient glenohumeral internal rotation and total rotation, deficit of strength in the rotator cuff, and scapular dyskinesia. The most common biomechanics adaptation is posterior stiffness of the shoulder, causing a decreased horizontal adduction of the shoulder and reduced mobility in internal rotation, causing capsular tightness, and muscle spasm. In the same way, posterior shoulder stiffness, therefore, has been suggested to be a causative or perpetuating factor in shoulder impingement and labral pathology [7].

Soft tissue mobilization techniques [8] can increase internal rotation and horizontal adduction movements of the shoulder [9] and the range of knee and hip motion affecting quadriceps and hamstrings [10,11]. It has been reported [12] that these techniques can reduce rotator cuff stiffness to improve the range of motion, as well as reduce the pain threshold of an active and musculoskeletal movement [9,13]. In the same way, soft tissue mobilization techniques may have an inhibitory effect on hyperactive muscles, thus favoring intermuscular balance [14].

The current literature provides support for instrument-assisted soft tissue mobilization (IASTM) in improving the range of motion (ROM) in uninjured individuals as well as pain and patient-reported function (or both) in injured patients [15]. Horizontal adduction shoulder stretch or post-isometric cross-body stretch can improve the range of motion in horizontal adduction and the glenohumeral internal rotation [16], by decreasing the posterior stiffness of the shoulder [17]. Moreover, IASTM appeared to be effective in yielding short-term improvements in shoulder horizontal adduction and internal rotation among uninjured participants [12]. It is recommended to perform stretching by stabilizing the scapula to decrease infraspinatus stiffness and avoid subacromial impingement [18].

The hypothesis of this study was that an intervention using instrument-assisted soft tissue mobilization and horizontal adduction shoulder produces improvements in the mobility of internal rotation and horizontal adduction of the shoulder, as well as the perception of stretching of the back of the shoulder in CrossFitters.

The aim of this study is to evaluate the effectiveness of a physical therapy intervention through instrument-assisted soft tissue mobilization and horizontal adduction shoulder stretches in CrossFitters aged from 18 to 40 years.

## 2. Materials and Methods 

### 2.1. Study Design and Approvals

Randomized, single-blind pilot study was conducted with CrossFit athletes from the gym Acero CrossFit, located in the city of Toledo (Spain). The study compared clinical outcome after instrument-assisted soft tissue mobilization techniques and post-isometric horizontal adduction stretches or underwent soft tissue mobilization with 21 athletes randomised to each intervention type.

The study was registered at www.clinicaltrials.gov (NCT03830346.). This study has been approved by the Research Committee of the European University of Madrid (registration no.: CIPI/18/033). Prior to the commencement of the study, all the participants selected signed an informed consent document, as defined by the Helsinki Rules.

### 2.2. Study Population

We calculated the sample size needed for this study (effect size = 0.25 (medium), α error = 0.05, power = 0.8) using the G*power software (Version 3.1., Heinrich Heine University, Duesseldorf, Germany). The effect size used herein was in accordance with a previous study [19]. The results showed that 18 participants were required. Given the likelihood of dropouts during the study, a total of 24 participants were recruited, of which 21 met the selection criteria and were included in the study. The athletes were invited (in February) to participate in the study. The study period was from January to June 2018.

The inclusion criteria to participate in the study were as follows: participants of both sexes, being regular CrossFitters (workouts at least two days a week), and in the age range of 18 to 40 years. On the other hand, participants excluded were those who had suffered a shoulder injury in the 3 months prior to the study, had undergone shoulder surgery in the previous six months, had a non-attendance rate of over 15% of the intervention sessions (2 sessions), and had not signed the informed consent document.

### 2.3. Randomisation 

Participants who met the selection criteria, and after signing the informed consent document, were randomly assigned by the opaque envelope system to each study group: experimental group (*n* = 11) and control (*n* = 10). Participants were randomly allocated by a person not involved in the study.

### 2.4. Outcome Evaluation

Three dependent variables were evaluated: mobility of internal rotation and horizontal adduction of the shoulder, and the perception of stretching of the back of the shoulder in each movement.

The assessment of the internal rotation and horizontal adduction of the shoulder was performed according to the protocol described by Laudner et al. [20]. In order to measure the internal rotation of the shoulder, the patient was placed in the supine position on the stretcher with the arm to be assessed at 90 degrees shoulder adduction, 90 degrees elbow flexion, and with the elbow at the height of the acromion with a towel. By stabilizing the scapula at the acromion, the shoulder was taken at a maximum range of internal rotation. The range of motion was measured with a digital inclinometer, model Tacklife MDP01. The angle of the edge of the ulna coincided with a line perpendicular to the stretcher. For horizontal adduction, the arm was placed in the same initial position in neutral rotation and, while stabilizing the lateral edge of the scapula, the arm was adducted to its maximum range of motion. The angle between the line of the ventral edge of the humerus and a line perpendicular to the stretcher was measured with the inclinometer. Intraclass correlation coefficient and standard error of measurement values were 0.93 and 1.6° for passive Glenohumeral (GH) horizontal adduction ROM and 0.98 and 2.0° for internal rotation ROM, respectively.

To assess the perception of stretch, the scale described by Park et al. [21] was used. This 11-point scale evaluates the discomfort from least to most, asking each participant to define the level of discomfort in the back of the shoulder in the maximum range of motion of internal rotation and horizontal adduction of the shoulder. Intraclass correlation coefficient for this scale was 0.97 (95% confidence interval (CI) = 0.96 to 0.98).

The main anthropometric independent variables were collected (height, weight, and body mass index), as well as sociodemographic variables (sex, age, profession, experience, weekly training sessions, duration of training, competition, and so on).

Three assessments were carried out in this study: prior to intervention (T0), following intervention (T1), and after a 4-week follow-up period (T2). Another physical therapist oversaw conducting the three study assessments, blinded with respect to participant allocation to each study group. All assessments were carried out following the same protocol and under the same conditions.

### 2.5. Intervention

Each session lasted 2 to 5 min, 2 days a week, over a period of 4 weeks, prior to each workout. In the experimental group, instrument-assisted soft tissue mobilization techniques and post-isometric horizontal adduction stretches were performed, while the control group only underwent soft tissue mobilization.

The soft tissue mobilization techniques were applied with the participant in prone position, as described by Laudner et al. [20]. The technique lasted 20 s in a parallel direction and 20 s in a perpendicular direction on the posterior shoulder and scapula muscles. While the dominant hand was used to hold the instrument, the other hand was used to tighten the skin medially to ensure an even area of treatment (Figure 1).

The post-isometric horizontal adduction stretch was carried out according to the protocol by Moore et al. [22] in the movement evaluation position described by Laudner et al. [20] with the participant in the supine position, passively adducting the arm horizontally until the first motion barrier and performing active horizontal adduction for 5 s at 25% of force. The arm was then taken to the new motion barrier, repeating this process three times.

### 2.6. Statistics

Sample distribution analysis was performed using the Shapiro–Wilk test. The differences between the three assessments were analyzed, in each group, for the different variables using the non-parametric Wilcoxon test. An analysis of variance (ANOVA) of repeated measures was carried out to compare the experimental and control groups at the three assessment times: baseline (T0), posttreatment (T1), and follow-up (T2). The results of the F test depend on whether the Mauchly spherical test was significant or not. If significant, the Greenhouse–Geisser correction was used. Bonferroni correction has been applied to control the error rate of the significance level. When the interaction was significant, pairwise comparison tests were performed on the group. The partial eta-squared value was calculated as an indicator of effect size (classified as small 0.01, medium 0.06, and large 0.14) [23]. An analysis by intention to treat was conducted. The level of significance of the study was estimated at 95%.

## 3. Results

During the study and follow-up, none of the participants included in the experimental group (*n* = 11) or control group (*n* = 10) dropped out. Figure 2 shows the flowchart of the research study.

The average age of the 21 participants included in the study was 30.81 years (SD: 5.35), with an average height of 178 (SD: 7.93) cm, an average weight of 82.69 (SD: 10.82) kg, and a mean body mass index of 25.98 (SD: 3.04) kg/m^2^. The mean of weekly training sessions was 4.1, with an average session duration of 82 min, and the length of time since initiating in the practice of CrossFit being 29.38 months on average. Furthermore, 90.5% of participants were males, and only 28.6% of the participants had ever competed. Although the pretreatment assessment revealed no differences (*p* > 0.05) in anthropometric variables and internal rotation and horizontal adduction movements, all other independent variables and the measurements of perception of stretching of all movements showed differences (*p* < 0.05) between the two groups. The description of the whole sample, and according to the group, is shown in Table 1.

Table 2 shows the statistical analysis of the dependent variables of the study at baseline, post-treatment, and follow-up assessment. The experimental group revealed changes in all variables (*p* < 0.001) after the intervention. When comparing T0 and T2 assessments, we found improvements in all variables (*p* < 0.01) The calculation of the effect size in the post-treatment results produced high values (d > 0.80) in all variables, except perception of left internal rotation (d = −0.58) and perception of left horizontal adduction (d = −0.80). Similarly, the effect size obtained after follow-up period was high (d > 0.80) in range of motion variables, and moderate (range: −0.58 to −0.75) in the other variables.

Differences were found (*p* < 0.01) in the control group between T0 and T1 assessments in right horizontal adduction and left internal rotation. When comparing T0 and T2 assessments, we found improvements in five variables: right internal rotation, perception of right internal rotation, right horizontal adduction left internal rotation, and left horizontal adduction (*p* < 0.01). The effect size obtained after the follow-up period was high (d > 0.80) in range of motion variables and perception of right internal rotation (d = 1.67). Table 3 shows the main statistics of the three assessments performed in the two groups.

There were differences between the three evaluations in the perception of stretch in all motions; however, no differences were found in the group interaction in terms of range of motion. No significant difference was reported in dependent variables, upon comparing the three assessments (T0, T1, and T2). Table 4 shows the results of the repeated measures analysis including baseline (T0), T1, and T2 assessments.

## 4. Discussion

The study examined the effectiveness of instrument-assisted soft tissue mobilization and post-isometric horizontal adduction stretches in CrossFitters. The results of this study support the assumption that this intervention may have a positive effect on range of motion and perception of stretch. The high effect size found after post-treatment and follow-up assessments indicates a high power of the results. These data suggest that the application of instrument-assisted soft tissue mobilization techniques and post-isometric horizontal adduction stretches can generate improvements after 4 weeks of intervention that are maintained after 4 weeks of follow-up.

CrossFit is a highly popular conditioning program combining elements of strength, coordination, balance, and mobility. It represents one of the most common examples of high-intensity interval training [24] However, there are no clinical studies on CrossFitters comparable to our study. This absence of scientific evidence complicates the possibility of comparing results in similar samples, although the techniques used have already been used in other studies. Laudner et al. [20] observed improvements in baseball players in range of motion in horizontal adduction and internal rotation of the dominant shoulder, through a single application of instrument-assisted mobilization, without assessing whether the improvements were maintained over time. Despite involving different populations and not being comparable, our study included 39 shoulders, observing an improvement that is maintained even after 4 weeks. 

The literature suggests that IASTM is effective in increasing acute shoulder ROM in overhead athletes with asymptomatic ROM deficiency. The lack of a standardized IASTM treatment protocol in the current research presents a limitation to utilize it in clinical practice [12]. McMurray et al. [25] reported that treatment sessions usually last approximately 5–6 min per treatment region.

McClure et al. [26] noted, after 4 weeks of intervention, how cross-body stretches in asymptomatic participants were more effective than sleeper stretches commonly used in the improvement of horizontal adduction and internal rotation of the shoulder. Similarly, Manske et al. [27] showed how cross-body stretches with joint mobilization were more effective than stretching alone in the improvement of range of motion in internal rotation in asymptomatic participants. Our study includes this technique, where the interventions were carried out before the training session, with the goal of providing CrossFitters with that range of motion for their exercises, and thus be able to perpetuate the effect of the intervention. Moreover, the improvement observed in the study prior [27] is maintained in our results in terms of shoulder ROM.

Bailey et al. [28] evaluated the effectiveness of cross-body and sleeper self-stretches both alone and combined with instrument-assisted soft tissue mobilization, with each intervention lasting 4 min, noting how the group treated with soft tissue mobilization improved internal rotation and horizontal adduction of the shoulder. Our study found improvements in both movements, with special significance for horizontal adduction. However, our intervention protocol included shorter application times. Thus, it can be established that shorter treatment times, as shown in our study, produce improvements after 4 weeks of intervention, and that these are maintained over time.

The study findings show that the time needed for treatment in shoulder movement restriction can be reduced. By applying a soft tissue mobilization technique for 40 s, instead of over a minute and a half stretch per shoulder, the time of treatment is reduced by almost three quarters. The use of this protocol as a preventive measure of shoulder injuries would be desirable in CrossFit participants favoring an improved mobility.

Study limitations include the low sample size, although no participant dropped out of the study. To address this limitation, the values of effect size were calculated to observe the statistical power of the results in our sample. A larger team of researchers would have been desirable to facilitate the process of intervention and evaluation. Finally, the completion of the three assessments on the same day of the week and at the same time could provide different results to those found in this study.

The current review highlights three important factors associated with injury incidence and incidence rates in CrossFit: training frequency, duration of CrossFit experience, and individuals that compete in CrossFit competitions [29]. Future research should include a larger sample size, with the sample being homogeneous. In addition, more dependent variables such as muscle strength of the shoulder should be assessed.

## 5. Conclusions

The instrument-assisted soft tissue mobilization technique with post-isometric horizontal adduction stretches may improve the range of motion of the shoulder. These improvements can be maintained for up to four weeks. A protocol that includes an instrument-assisted soft tissue mobilization technique can improve horizontal adduction and internal rotation. Better results would potentially be attainable by adding the horizontal adduction shoulder stretch technique. Instrument-assisted soft tissue mobilization has no adverse effects or complications in asymptomatic participants.

## Figures and Tables

**Figure 1 ijerph-18-00575-f001:**
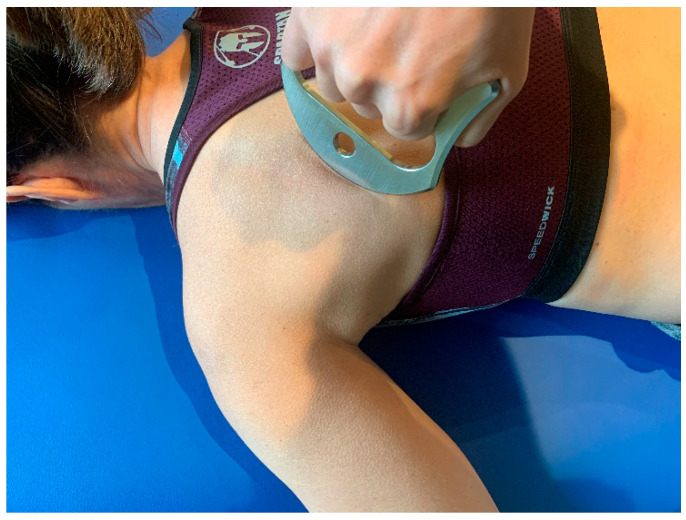
Soft tissue mobilization techniques in prone position.

**Figure 2 ijerph-18-00575-f002:**
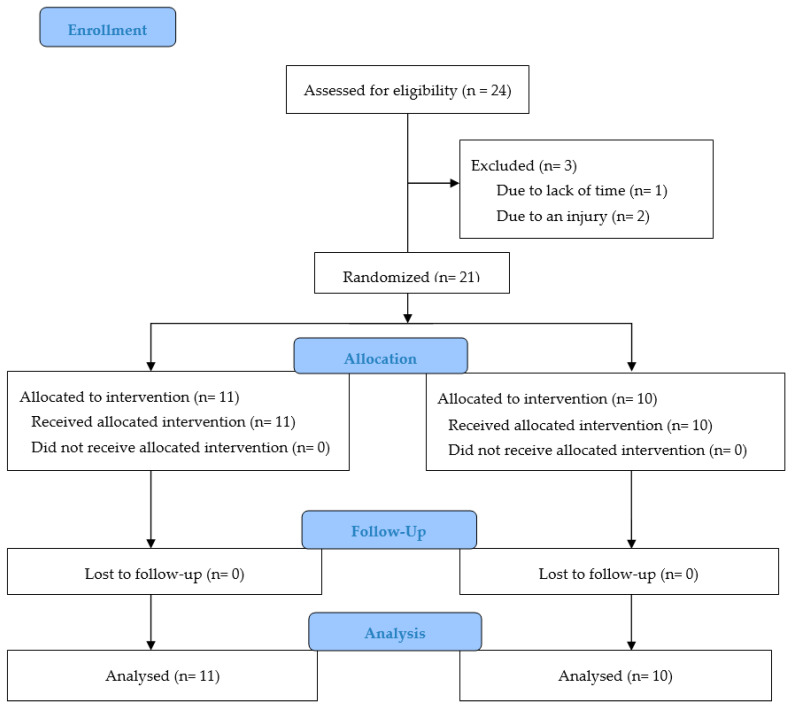
CONSORT 2010 flow diagram.

**Table 1 ijerph-18-00575-t001:** Descriptive characteristics of all patients mean (and standard deviation) at baseline and in each group of the study.

Psychometric Variables	All Sample		Experimental Group		Control Group		Sig.
Age (years)	30.81 (5.35)		31.45 (6.02)		30.10 (4.72)		0.16 ^a^
Height (cm)	178.33 (7.93)		178.27 (9.07)		175.36 (7.68)		0.14 ^a^
Weight (kg)	82.69 (10.82)		81.82 (12.18)		70.93 (11.81)		0.48 ^a^
Body mass index (kg/m^2^)	25.98 (3.04)		25.60 (2.29)		22.92 (2.44)		0.06 ^a^
Clinical variables							
Time practicing CrossFit (months) *	29.38 (20.69)		41 (19.32)		16.6 (13.81)		0.02 ^a^
Training per week (days) *	4.1 (1.22)		4.73 (0.90)		3.40 (1.17)		0.03 ^a^
Time per training (minutes) *	82.14 (28.31)		91.36 (28.81)		72 (25.29)		0.00 ^a^
Sociodemographic variables	*n*	%	*n*	%	*n*	%	
Gender (Male/Female)	19/2	90.5/9.5	10/1	90.9/9.1	9/1	90/10	0.78 ^b^
Participation in competition (Yes/No) *	6/15	28.6/71.4	4/7	36.4/63.6	2/8	20/80	0.04 ^b^

M: mean; SD: standard deviation; *n*: number of participants; %: percentage; Sig.: significance. ^a^ Shapiro–Wilks test. ^b^ Fisher exact test. * Significant difference (*p* < 0.05).

**Table 2 ijerph-18-00575-t002:** Statistical analysis and median (and interquartile range) of the dependent variables of the study at baseline, post-treatment, and follow-up assessment.

Variables	Experimental Group			Control Group		
	T0	T1	T2	T0	T1	T2
Right internal rotation	36.4 (18.0)	51.1 (11.8)	48.5 (12.0)	38.65 (15.8)	44.00 (16.4)	55.90 (10.8)
Perception of right internal rotation	3.00 (2.0)	2.00 (1.0)	2.00 (1.0)	1.00 (1.5)	2.00 (2.2)	3.00 (2.7)
Right horizontal adduction	12.2 (12.0)	19.2 (4.0)	19.1 (5.0)	12.9 (10.2)	16.85 (7.1)	16.80 (17.1)
Perception of right horizontal adduction	3.00 (2.0)	2.00 (2.0)	2.00 (1.0)	2.00 (3.0)	2.00 (0.25)	3.00 (1.0)
Left internal rotation	38.5 (13.1)	43.9 (15.8)	54.8 (5.5)	44.15 (23.4)	50.45 (10.1)	58.35 (12.0)
Perception of left internal rotation	3.00 (4.0)	2.00 (3.0)	2.00 (4.0)	2.00 (2.25)	3.00 (1.5)	2.50 (2.2)
Left horizontal adduction	15.7 (6.9)	20.30 (4.8)	21.5 (9.3)	11.90 (5.9)	16.05 (9.1)	22.45 (13.4)
Perception of left horizontal adduction	4.00 (1.0)	2.00 (2.0)	2.00 (2.0)	2.50 (3.0)	2.50 (3.0)	2.50 (3.0)

Outcome measures at the baseline (T0), after the four-week period of soft tissue mobilization and control interventions (T1), and after a further four weeks as follow-up (T2).

**Table 3 ijerph-18-00575-t003:** Mean difference and changes (and effect size) after post-treatment and follow-up period of the dependent variables of the study with non-parametric Wilcoxon test.

Variables	Experimental Group		Control Group	
	T0–T1	T0–T2	T0–T1	T0–T2
Right internal rotation	−13.87 (1.07) *	−16.58 (1.28) *	−3.88 (0.33)	−14.90 (1.29) *
Perception of right internal rotation	1.27 (−0.81) **	1.00 (−0.64) *	−0.90 (0.83)	−1.80 (1.67) *
Right horizontal adduction	−6.79 (1.21) **	−6.93 (1.23) **	−3.16 (0.46) *	−6.13 (0.89) *
Perception of right horizontal adduction	2.00 (−0.86) **	1.45 (−0.62) **	0.10 (−0.06)	−0.20 (0.12)
Left internal rotation	−12.05 (1.47) **	−18.25 (−6.56) **	−8.77 (0.54) *	−13.48 (0.83) *
Perception of left internal rotation	1.36 (−0.58) **	1.36 (−0.58) **	−0.800 (0.38)	−0.900 (0.42)
Left horizontal adduction	−4.40 (1.20) **	−6.03 (1.64) **	−4.01 (0.65)	−9.27 (1.51) *
Perception of left horizontal adduction	1.63 (−0.80) **	1.54 (−0.75) **	0.10 (−0.04)	−0.30 (0.14)

Outcome measures at the baseline (T0), after the four-week period of treatment and control interventions (T1), and after a further four weeks as follow-up (T2). * Significant difference between improvements of the study groups (*p* < 0.01). ** Significant difference between improvements of the study groups (*p* < 0.001).

**Table 4 ijerph-18-00575-t004:** Statistical analysis of repeated measures of the dependent variables in the three study assessments.

Variables	Intra-Group Effect			Inter-Group Effect		
	F	Sig.	η^2^_p_	F	Sig.	η^2^_p_
Right internal rotation ^a^	0.53	0.59	0.02	0.87	0.36	0.04
Perception of right internal rotation ^a^	8.38	0.00 *	0.30	0.08	0.77	0.01
Right horizontal adduction ^a^	2.02	0.15	0.09	0.20	0.65	0.01
Perception of right horizontal adduction ^a^	8.49	0.01 *	0.30	0.13	0.71	0.01
Left internal rotation ^a^	1.46	0.24	0.07	0.68	0.41	0.03
Perception of left internal rotation ^a^	3.70	0.04 *	0.16	0.02	0.86	0.01
Left horizontal adduction ^a^	1.56	0.22	0.07	0.01	0.89	0.00
Perception of left horizontal adduction	4.98	0.01 *	0.20	1.11	0.30	0.05

Sig.: significance; η^2^_p_: partial Eta-squared. ^a^ the df corresponds to Greenhouse–Geisser test. * Interaction with the group (*p* < 0.05).

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
