# Peer review of "Soft Tissue Mobilization and Stretching for Shoulder in CrossFitters: A Randomized Pilot Study"

_ijerph, 2021, doi:10.3390/ijerph18020575_

Round 1

Reviewer 1 Report

Dear Authors,

I have read the manuscript with great interest. I believe that it can be improved upon completion of a small revision.

-Most importantly, do authors consider including a new figure/schematic drawing explaining the 'instrument assisted soft tissue mobilization'? I believe this can help the readers who are not experts in the field.

-The quality/resolution of Figure 1 is low. Can you please improve it?

-One wonder if the authors detected any age-related differences in the study group?

Thanks.

Author Response

Dear Authors,

I have read the manuscript with great interest. I believe that it can be improved upon completion of a small revision.

-Most importantly, do authors consider including a new figure/schematic drawing explaining the 'instrument assisted soft tissue mobilization'? I believe this can help the readers who are not experts in the field. We have added a new figure that contains the image of the technique used in the study, to make it easier for the reader to understand.

-The quality/resolution of Figure 1 is low. Can you please improve it? We have reintroduced figure 1 to make it look better.

-One wonder if the authors detected any age-related differences in the study group? Although it was not the objective of the study, we did not find significant differences regarding the age variable. Therefore, we do not include the correlation analysis in the text or in tables.

Reviewer 2 Report

  • The authors examined soft tissue mobilization and stretching for shoulder in CrossFitters. This study is interesting, but there are some concerns.
  • There are many grammatical issues throughout the text that need to be fixed.
  • While research “subject” is the more traditional of the two terms, there has been a shift over the past 25 years or so to use research “participant” when referring to individuals who take part in research, because, many argue, it is more respectful of research volunteers.

Abstract

  • The abstract needs to contain anthropometric characteristics of participants such as age, height, weight, etc.

Introduction

  • Line 43: Use the “%” symbol for 13.1. Consistency makes the manuscript look more polished and professional.
  • Lines 71-72: The following sentence seems incomplete and needs to be more elaborated:

“In the same way, shoulder impingement and the glenoid labrum pathology sometimes develop [6].”

  • Lines 81-87: The following sentence is long and unclear. You need to split it into two or more sentences and make it more straightforward:

“Also, stretches in adduction for 4 weeks can improve internal shoulder rotation, and together with shoulder mobilization, better results are achieved even after follow-up [21], while it is recommended to perform stretching by stabilizing the scapula to decrease infraspinatus stiffness and avoid subacromial impingement [22,23].”

  • The study “Hypotheses” need to be included in the “Introduction”.

Methods

  • Day-to-day test reliability, CV range, and intraclass correlation coefficients for the assessments needs to be included for ALL the assessments.
  • Suggestion: Add a schematic representation of the study procedures to the “Methods” section.

Results

  • Figure 1 is vague. Please consider fixing that. Also, this figure needs to be moved to the “Methods” section.
  • Line 179: Write “90.5% in letters since it’s at the beginning of the sentence.

Author Response

The authors examined soft tissue mobilization and stretching for shoulder in CrossFitters. This study is interesting, but there are some concerns.

There are many grammatical issues throughout the text that need to be fixed. The text has been checked by a professional translator.

While research “subject” is the more traditional of the two terms, there has been a shift over the past 25 years or so to use research “participant” when referring to individuals who take part in research, because, many argue, it is more respectful of research volunteers. As the reviewer indicates, we have changed the word subject to participant in the text, to adapt it to a language more in line with the scientific method.

Abstract

  • The abstract needs to contain anthropometric characteristics of participants such as age, height, weight, etc. As the reviewer points out, we have included the main anthropometric values of the participants in the abstract.

Introduction

  • Line 43: Use the “%” symbol for 13.1. Consistency makes the manuscript look more polished and professional. As data on the prevalence of injuries is offered in the text, we cannot indicate percentages, since it is data for hours of training (1 injuries per 1000 hours of training). We hope that the reviewer understands that a data in% offers a wrong value, which is not what we intend to offer the reader.
  • Lines 71-72: The following sentence seems incomplete and needs to be more elaborated: “In the same way, shoulder impingement and the glenoid labrum pathology sometimes develop [6].” As the reviewer points out, the sentence has been corrected to facilitate understanding: “Posterior shoulder stiffness, therefore, has been suggested to be a causative or perpetuating factor in shoulder impingement and labral pathology”.
  • Lines 81-87: The following sentence is long and unclear. You need to split it into two or more sentences and make it more straightforward: “Also, stretches in adduction can improve internal shoulder rotation, together with shoulder mobilization, better results are achieved [21]. It is recommended to perform stretching by stabilizing the scapula to decrease infraspinatus stiffness and avoid subacromial impingement [22,23].” We have shortened the sentence, separating it into two sentences to make it easier to read.
  • The study “Hypotheses” need to be included in the “Introduction”. As the reviewer points out, we have included the study hypothesis.

Methods

  • Day-to-day test reliability, CV range, and intraclass correlation coefficients for the assessments needs to be included for ALL the assessments. We have included the Intraclass correlation coefficient values that exist in the literature, as well as the standard error and the 95% CI that in both scales is offered in the literature.
  • Suggestion: Add a schematic representation of the study procedures to the “Methods” section. To facilitate the reader's understanding, and as indicated by both reviewers, we have added a figure of the intervention to facilitate understanding.

Results

  • Figure 1 is vague. Please consider fixing that. Also, this figure needs to be moved to the “Methods” section. We have reintroduced figure 1 to make it look better. The other reviewer has not indicated the change of location from Figure 1 to the Methods We await your instructions.
  • Line 179: Write “90.5% in letters since it’s at the beginning of the sentence. We have added the numerical value in letters (Ninety point five percent of participants…) as indicated by the reviewer.